# Iron Single Atoms Anchored on Carbon Matrix/g-C_3_N_4_ Hybrid Supports by Single-Atom Migration-Trapping Based on MOF Pyrolysis

**DOI:** 10.3390/nano12091416

**Published:** 2022-04-20

**Authors:** Yining Jia, Rong Huang, Ruijuan Qi

**Affiliations:** 1Key Laboratory of Polar Materials and Devices, Ministry of Education, School of Physics and Electronic Science, East China Normal University, Shanghai 200241, China; 51191213010@stu.ecnu.edu.cn; 2Collaborative Innovation Center of Extreme Optics, Shanxi University, Taiyuan 030006, China

**Keywords:** atom migration-trapping, single-atom catalysts, metal–organic frameworks

## Abstract

Numerous efforts have been devoted to realizing the high loading and full utilization of single-atom catalysts (SACs). As one of the representative methods, atom migration-trapping (AMT) is a top-down strategy that converts a certain volume of metal nanoparticles (NPs) or metal-based precursors into mobile metal species at high temperature, which can then be trapped by suitable supports. In this study, high-loading iron single atoms anchored onto carbon matrix/g-C_3_N_4_ hybrid supports were obtained through a single-atom migration-trapping method based on metal–organic framework (MOF) pyrolysis. It is confirmed, by high-angle annular dark field scanning transmission electron microscopy (HAADF-STEM), X-ray absorption near-edge structure (XANES) and extended X-ray absorption fine structure (EXAFS), that the Fe(acac)_3_ precursor is reduced to Fe single atoms (SAs), which are not only anchored onto the original N-doped carbon (NC), but also onto g-C_3_N_4_, with an Fe-N coordination bond. Further electrochemical results reveal that Fe-C_3_N_4_-0.075 possesses a better half-wave potential of 0.846 V and onset potential of 0.96 V compared to Fe-N-C, the product obtained after pyrolysis of Fe(acac)_3_@ZIF-8. As opposed to SAs prepared by the pyrolysis process only, SAs prepared by AMT are commonly anchored onto the surface of the supports, which is a simple and effective way to make full use of the source metal and prepare SACs with higher exposing active sites.

## 1. Introduction

Since Zhang’s group proposed the pioneering work of Pt single-atom catalysts (SACs) fabricated on metal oxides for efficient CO oxidation in 2011 [1], various synthesis methods have been proposed for the preparation of SACs, including immersion methods [2,3,4], co-precipitation [1], atomic layer deposition [5,6,7], electrochemical deposition [8,9,10], photochemical reactions [11,12,13], high-temperature atom trapping methods [14], chemical vapor deposition [15] and pyrolysis [16,17,18]. Among them, pyrolysis is an extensively studied process, in which SACs are formed by the thermal decomposition of suitable precursors at high temperatures, under a controlled atmosphere (i.e., inert gases or NH_3_). 

Metal–organic frameworks (MOFs) are ideal precursors for the synthesis of SACs via pyrolysis, due to their abundant and uniformly distributed organic ligands, which can be converted into carbon supports with abundant N sites and transition metal sites after pyrolysis [19,20,21], forming a firm metal–nitrogen coordination bond, which greatly improves the thermal and chemical stability of SACs, preventing the aggregation of single atoms (SAs). However, there are some disadvantages of this method, such as low loading of SAs, due to limited active sites on the finite supports (e.g., carbon), which leads to waste of the source metal. In contrast, atom migration-trapping (AMT) is a top-down strategy that uses metal nanoparticles (NPs)/bulk metal-based materials as precursors to obtain different mobile metal species at high temperature; then, by binding to coordination elements in appropriate supports, the mobile metal species can be trapped and eventually formed as stable SAs [22]. The high temperature required for AMT ensures that the resulting SAs will only occupy the most stable binding sites, exhibiting good sintering resistance [22]. The AMT method must meet two prerequisites: the generation of mobile metal species and suitable supports that can trap and stabilize the mobile species. Currently, the supports commonly used for AMT are reducible oxides or some 2D nanomaterials. Datye’s group has successively implemented AMT on Pd/CeO_2_ [23], Pd-La/Al_2_O_3_ [24] and Pt-Sn/CeO_2_ [25], which has verified the universality of this strategy on reducible oxide supports. Li et al. [26] took bulk platinum metal as the precursor, and ammonia produced by DCD pyrolysis had a strong coordination effect with Pt atoms. Mobile Pt(NH_3_)_x_ could be immobilized on the defective graphene surface to generate atomically dispersed Pt SAs/DG. Wu et al. [27] obtained Ni SAs by loading Ni NPs on defective N-doped carbon (NC) derived by ZIF-8 after pyrolysis under Ar at 900 °C.

Inspired by the above reports, we tried to obtain high-loading iron single atoms anchored onto carbon matrix/g-C_3_N_4_ hybrid supports by AMT based on MOF pyrolysis. Herein, Fe(acac)_3_@ZIF-8 and different molar amounts of g-C_3_N_4_ were mixed and pyrolyzed in N_2_ at 900 °C. Combined with scanning electron microscopy (SEM), transmission electron microscopy (TEM) and high-angle annular dark field scanning transmission electron microscopy (HAADF-STEM) techniques, we found that the content of g-C_3_N_4_ played a critical role in the morphology of the final products. With a higher content of g-C_3_N_4_, carbon matrices with obvious Fe NPs and plenty of carbon nanotubes (CNTs) were obtained. However, high-loading Fe SAs on carbon matrices and g-C_3_N_4_ hybrid supports could be obtained by tuning the content of g-C_3_N_4_. This indicates that, during the pyrolysis of the mixture, some part of Fe(acac)_3_ in the molecular cage of ZIF-8 can be reduced with the carbonization of the organic ligands at high temperature, and can be anchored onto NC to form isolated metal SAs; the other part can generate mobile metal species, which can be trapped and form isolated metal SAs, due to the chemical inertness and thermal stability of g-C_3_N_4_. Compared to SAs prepared by the pyrolysis process only, SAs prepared by AMT are commonly anchored onto the surface of the supports. We believe that this is a simple and effective way to make full use of the source metal and prepare SACs with more exposing active sites.

## 2. Materials and Methods

### 2.1. Preparation of Fe(acac)_3_@ZIF-8

A total of 0.6082 g Zn(NO_3_)_2_·6H_2_O (analytical grade, 99%, Aladdin, Shanghai, China) and 71.6 mg Fe(acac)_3_ (98%, Aladdin) were dissolved in 15 mL methanol (analytical grade, Sinopharm Chemical, Shanghai, China), and stirred for 15 min in beaker A. Next, 1.0604 g 2-methylimidazole (98%, Aladdin) was dissolved in 7.5 mL methanol and stirred for 15 min in beaker B. The solution in beaker B was subsequently added into beaker A with thorough stirring for 0.5 h at room temperature. Then, the mixed solution was transferred into a 45 mL Teflon-lined stainless-steel autoclave and kept at 90 °C for 1 h in an oven. After cooling to room temperature, the obtained product was separated by centrifugation and washed with anhydrous ethanol three times, and finally dried at 60 °C in a vacuum oven overnight.

### 2.2. Preparation of g-C_3_N_4_

A total of 10 g CN_2_H_4_O (Macklin Inc., Shanghai, China) was placed into a crucible and covered with a crucible lid. It was then placed into a Muffle furnace, heated to 550 °C in air and held for 3 h. The resulting product was washed with distilled water and dried. A light-yellow powder was obtained.

### 2.3. Pyrolysis of Fe(acac)_3_@ZIF-8 and g-C_3_N_4_

A total of 100 mg Fe(acac)_3_@ZIF-8 and 3 mg g-C_3_N_4_ were mixed uniformly and transferred into a ceramic boat. Then, the boat with the sample was heated to 900 °C at a heating rate of 5 °C min^−1^ and kept at 900 °C for 3 h under flowing N_2_. The final black powder was collected after natural cooling and was ready for subsequent characterization without further treatment. The corresponding product was labeled as Fe-C_3_N_4_-0.075. Fe-C_3_N_4_-0.12 was prepared similarly to Fe-C_3_N_4_-0.075, except that the amount of g-C_3_N_4_ was changed to 5 mg.

The product obtained from the pyrolysis of Fe(acac)_3_@ZIF-8 under the same conditions was named Fe-N-C.

### 2.4. Characterization

The morphology of the samples was characterized by SEM (Gemini 450, ZEISS, Jena, Germany) with an acceleration voltage of 5 kV. The TEM images and element mappings were acquired at 200 kV using a JEM-2100F (JEOL, Tokyo, Japan), equipped with an X-ray energy dispersive spectrometer (EDS: X-Max 80T, Oxford, UK) for chemical composition analysis. EDS elemental maps were taken in HAADF-STEM mode. Atomic resolution analyses were performed on an aberration-corrected scanning transmission electron microscope (AC-STEM, Grand ARM300F, JEOL, Tokyo, Japan). The X-ray absorption spectroscopy (XAS) measurements were carried out at the 1W2B beamline at the Beijing Synchrotron Radiation Facility (BSRF), China. The EXAFS data were processed according to the standard procedures using the ATHENA module implemented in the IFEFFIT software packages (version 0.9.25, Chicago, IL, USA).

### 2.5. Electrochemical Measurements

Commercial Pt/C (20 wt.%) was purchased from Alfa Aesar. A total of 2 mg as-prepared catalyst was dispersed in a mixture of 50 μL Nafion (5 wt.%), 633 μL isopropyl alcohol and 317 μL pure water, and sonicated for about an hour to form a homogeneous catalyst ink. A 24.7 μL ink was dropped onto the polished glassy carbon disk, in order to yield a catalyst loading of 0.2 mg cm^−2^, and dried in air at room temperature.

All the electrocatalytic performance measurements were carried out on a CHI 760E Electrochemical Workstation (Shanghai Chenhua, Shanghai, China). The oxygen reduction performance analysis was conducted using a three-electrode system at room temperature. A platinum wire and a saturated calomel electrode (*SCE*) were used as a counter electrode and reference electrode, respectively. The glassy carbon electrode was employed as the working electrode. All potential values were calibrated to the reversible hydrogen potential (*E_RHE_*), based on the Nernst equation of
$\;E_{RHE}=E_{SCE}+0.2415+0.0591\times \mathrm{pH}$.

The oxygen reduction reaction (ORR) test was performed in O_2_/N_2_-saturated 0.1 M KOH electrolyte at room temperature. Cyclic voltammetry (CV) experiments were conducted with a scan rate of 50 mV s^−1^ in the potential range of 0–1.2 V. Then, rotating ring-disk electrode (RRDE) measurements were conducted by Beijing Synchrotron Radiation Facility (LSV) at 1600 rpm with a scan rate of 5 mV s^−1^.

## 3. Results and Discussion

The morphologies of Fe(acac)_3_@ZIF-8 and g-C_3_N_4_ were characterized by SEM. As shown in Figure 1a, Fe(acac)_3_@ZIF-8 exhibits a rhombic dodecahedral morphology with a particle size of about 400 nm. In contrast, g-C_3_N_4_ shows a planar two-dimensional layered structure similar to graphene (Figure 1b), which can be easily distinguished from Fe(acac)_3_@ZIF-8. The SEM images of Fe-C_3_N_4_-0.075 and Fe-C_3_N_4_-0.12 are shown in Figure 1c,d. It was found that, with a lower content of g-C_3_N_4_, the pyrolyzed product still keeps a similar morphology to the raw mixture. However, when the molar ratio of g-C_3_N_4_ changes from 0.075 to 0.12, the morphology of the pyrolyzed product changes significantly. As shown in Figure 1d, rhombic dodecahedral particles with a wrinkled surface, surrounded by plenty of CNTs, can be observed.

TEM analyses further showed that Fe-C_3_N_4_-0.075 exhibited a clear mixture of two substances, with Fe(acac)_3_@ZIF-8-derived NC and 2D-layered g-C_3_N_4_ (Figure 2a). The HRTEM images in Figure 2b,c indicate that there are no obvious NPs on NC or g-C_3_N_4_, implying the possible formation of Fe SAs. The corresponding STEM-EDS elemental maps in Figure 2d–g demonstrate that the signals of Fe, N and C are uniformly dispersed on NC and g-C_3_N_4_. In contrast, the TEM image of Fe-C_3_N_4_-0.12 in Figure 2h clearly shows that many NPs and CNTs are distributed on the surface of the sample. However, no 2D-layered g-C_3_N_4_ can be observed. The HRTEM image in Figure 2i shows that the CNTs are bamboo-like. The HRTEM image in Figure 2j clearly shows the NPs and the layered structure of the carbon nanotube walls. The corresponding STEM-EDS elemental maps in Figure 2k–n demonstrate that the signals of N and C are uniformly dispersed on the surface of the sample, with obvious Fe NPs on NC. As reported, CNTs with metal NPs distributed at the end are easily formed by MOF pyrolysis with the presence of Fe or Co [28]. It seems that Fe or Co NPs can act as catalysts for the preparation of CNTs from amorphous carbon. In our work, we suppose that a higher content of g-C_3_N_4_ will restrict the volatilization or migration of Fe atoms during ZIF-8 pyrolysis, so that Fe atoms will agglomerate to form Fe NPs. The formed Fe NPs, in turn, act as catalysts for the transformation of 2D g-C_3_N_4_ to CNTs. To the best of our knowledge, this is the first report of the transformation of 2D g-C_3_N_4_ to CNTs under the possible catalysis effect of Fe, which may provide a new approach for the preparation of CNTs.

In addition, HAADF-STEM was used to further observe the atomic structure of Fe-C_3_N_4_-0.075. Since the atomic number (Z) of Fe atoms is much higher than that of C and N atoms [29], obvious bright dots, representing Fe SAs distributed on NC and g-C_3_N_4_, can be observed from the HAADF-STEM images (Figure 3b,c, corresponding to the areas marked in Figure 3a). For the region far away from the region of Figure 3a (Figure 3d), as shown in Figure 3e–f, a large number of Fe SAs anchored onto g-C_3_N_4_ can be observed. We can assume that, during the preparation of Fe-C_3_N_4_-0.075, the Fe source in Fe(acac)_3_@ZIF-8 formed mobile Fe species at high temperature, which were trapped and formed stable SAs on g-C_3_N_4_, successfully completing the “atomic migration-trapping”. It is well known that, for SACs, if the metallic catalysts are not fully encapsulated or simply anchored onto the surface of the carbon-based support, they cannot effectively prevent the leaching of metallic ions under harsh operating conditions. Moreover, fully encapsulating the metal catalyst in a thicker carbon matrix (similar to common MOF pyrolysis), but away from the surface, may prevent effective electron transfer between the catalyst and the reactants [30]. By combining MOF pyrolysis and the AMT method, we not only improve the utilization of the source metal, but also enhance the loading of SAs on the hybrid supports. As can be observed with Fe-C_3_N_4_-0.075, this method may liberate the SAs that cannot participate in the catalytic reaction due to being trapped inside the carbon-based supports, and load them onto the surfaces of other supports with more effective active sites.

For comparison, we also performed an HAADF-STEM analysis for Fe-C_3_N_4_-0.12. As shown in Figure 4a–d, it was found that, despite the generation of NPs, certain Fe SAs were generated on NC as well as the CNTs.

To further investigate the chemical state and coordination environment of the Fe center in Fe-C_3_N_4_-0.075, X-ray absorption near-edge structure (XANES) and extended X-ray absorption fine structure (EXAFS) were analyzed. As depicted in Figure 5a, the Fe K-edge XANES spectra of Fe-C_3_N_4_-0.075 are located between Fe foil and Fe_2_O_3_, indicating that the oxidation state of Fe species is between Fe^0^ and Fe^3+^. The EXAFS spectra of Fe-C_3_N_4_-0.075 (Figure 5b) showed a strong peak centered at 1.4 Å, which is mainly attributed to the scattering of Fe-N coordination. Furthermore, no scattering peaks derived from Fe-Fe coordination were observed in the Fe-C_3_N_4_-0.075 compared to Fe foil, demonstrating the mono dispersion of Fe species in the NC or g-C_3_N_4_. The coordination configuration of the Fe atom in Fe-C_3_N_4_-0.075 was further investigated by quantitative EXAFS curve fitting analyses (Figure 5c), which clearly revealed that the Fe center is coordinated with four N atoms at the first coordination shell. All the fitting results are generally consistent with the experimental data, from which the average coordination numbers of Fe-N were obtained to be 4.6 and the average bond length for Fe-N was 2.00 Å, respectively (Table 1). Additionally, the wavelet transform (WT, Figure 5d–f) results display only one WT intensity maximum at ≈3.8 Å^−1^, associated with an Fe-N pair. Compared with the WT plots of Fe foil, the WT signal related to the Fe-Fe contribution was not detected in the Fe-C_3_N_4_-0.075. These observations further confirm that the single Fe atoms simultaneously coordinate with N atoms, forming Fe-N bonds.

The construction of Fe-C_3_N_4_-0.075 was dedicated to possessing the ORR catalytic performance, which was investigated by CV and LSV in O_2_-saturated 0.1 M KOH solution, based on three electrode measurements under rotating disk conditions. A commercial 20 wt.% Pt/C catalyst and Fe-N-C were used as the reference for the performance comparison. All the given potentials refer to the reversible hydrogen electrode (RHE). As shown in Figure 6a, Fe-C_3_N_4_-0.075 exhibits a clear oxygen reduction peak, implying that it has oxygen reduction performance. Further, LSV tests were conducted. According to Figure 6b, the onset potential (E_onset_) with a value of about 0.96 V for Fe-C_3_N_4_-0.075 is inferior to that of Pt/C (1.01 V), but slightly better than that of Fe-N-C (0.955 V). Furthermore, Fe-C_3_N_4_-0.075 delivers a higher half-wave potential (E_1/2_ = 0.846 V) than that of Fe-N-C (0.83 V). This demonstrates that the addition of g-C_3_N_4_ does liberate Fe SAs originally hidden inside and anchor them by AMT, increasing the active sites involved in the reaction and leading to an improvement in the ORR performance.

## 4. Conclusions

In summary, high-loading iron single atoms anchored on hybrid supports were successfully prepared by combining MOF pyrolysis and the AMT method, using Fe(acac)_3_@ZIF-8 and g-C_3_N_4_ as raw materials. We not only improved the utilization of the source metal, but also enhanced the loading of SAs on the hybrid supports. According to our observations for Fe-C_3_N_4_-0.075, it may liberate the SAs trapped inside the carbon-based support that cannot participate in the catalytic reaction, and load them onto the surface of another support with more effective active sites. As is known, with the direct AMT of NPs, it is always difficult to ensure the complete utilization of metal, and the surface energy of NPs is much larger. However, the surface energy of the metal directly pyrolyzed from MOF is much smaller, making it easy to form mobile metal species, which is conducive to the full utilization of the source metal. In this way, Fe SAs anchored onto carbon matrix/g-C_3_N_4_ hybrid supports by single-atom migration-trapping based on MOF pyrolysis exhibit more active sites compared to the Fe-N-C, which is also reflected in the LSV curves. Although the ORR performance obtained in this study is not very high, it still provides a new idea for the preparation of SACs and lays the foundations upon which future performance improvements can be built.

## Figures and Tables

**Figure 1 nanomaterials-12-01416-f001:**
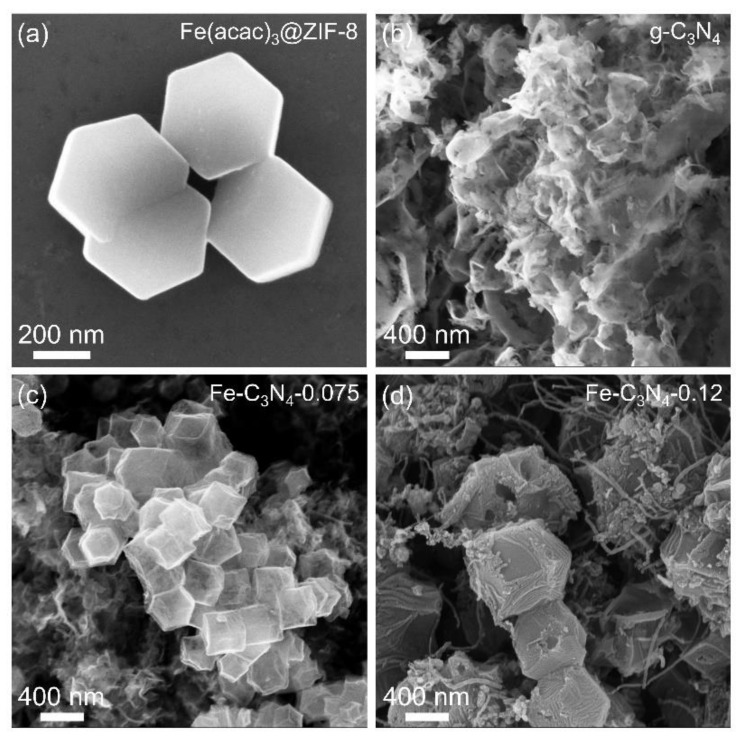
SEM images of (**a**) Fe(acac)_3_@ZIF-8, (**b**) g-C_3_N_4_, (**c**) Fe-C_3_N_4_-0.075 and (**d**) Fe-C_3_N_4_-0.12.

**Figure 2 nanomaterials-12-01416-f002:**
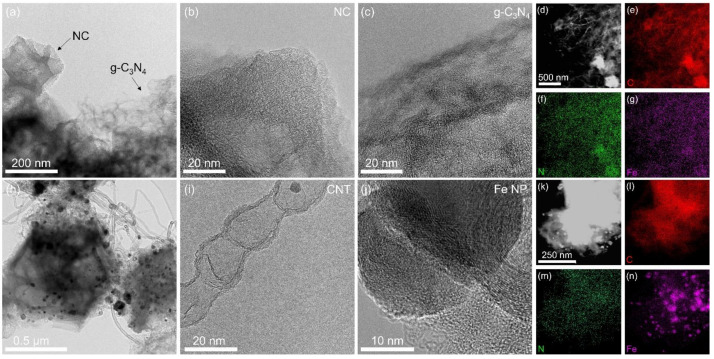
TEM image of (**a**) Fe-C_3_N_4_-0.075; (**b**) HRTEM image of NC and (**c**) HRTEM image of g-C_3_N_4_. (**d**–**g**) STEM-EDS elemental maps of C, Fe and N of Fe-C_3_N_4_-0.075 shown in (**d**). TEM image of (**h**) Fe-C_3_N_4_-0.12; (**i**) HRTEM image of CNTs and (**j**) HRTEM image of the nanoparticle. (**k**–**n**) STEM-EDS elemental maps of C, Fe and N of Fe-C_3_N_4_-0.12 shown in (**k**).

**Figure 3 nanomaterials-12-01416-f003:**
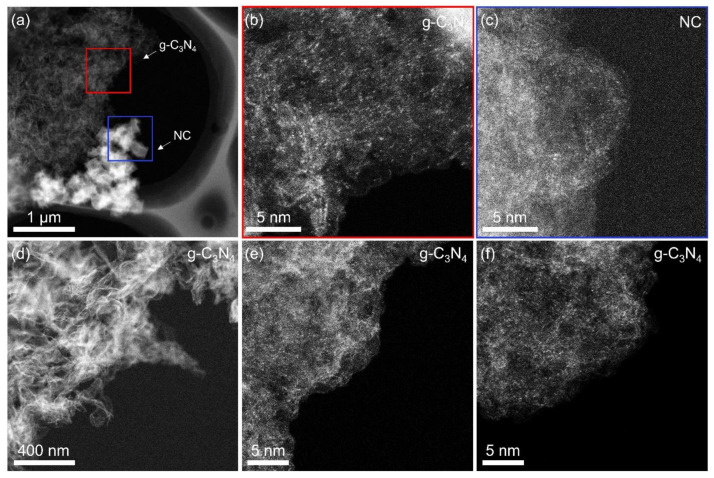
HAADF-STEM images of (**b**) g-C_3_N_4_ and (**c**) NC, corresponding to the red and blue areas in (**a**), respectively. (**d**–**f**) HAADF-STEM images of g-C_3_N_4_, where the region is far away from the region of (**a**).

**Figure 4 nanomaterials-12-01416-f004:**
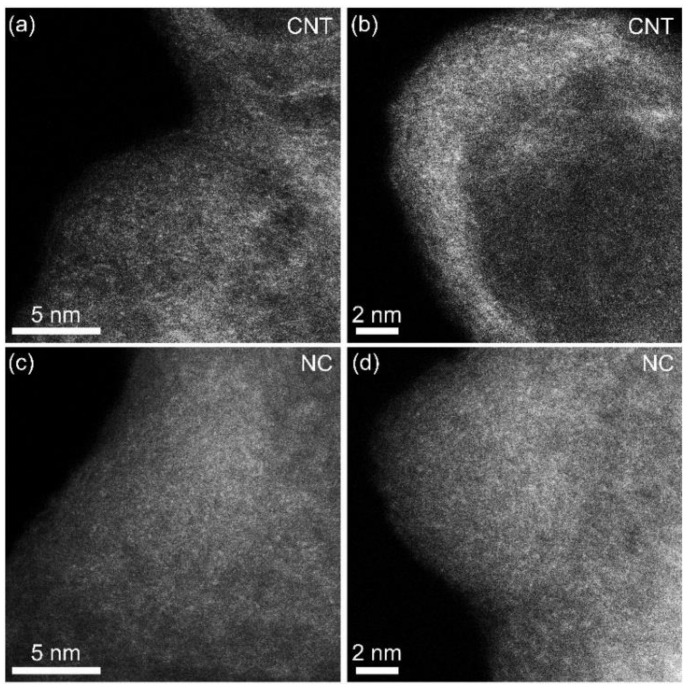
HAADF-STEM images of Fe-C_3_N_4_-0.12; (**a**) and (**b**) are of CNTs, and (**c**) and (**d**) are of NC.

**Figure 5 nanomaterials-12-01416-f005:**
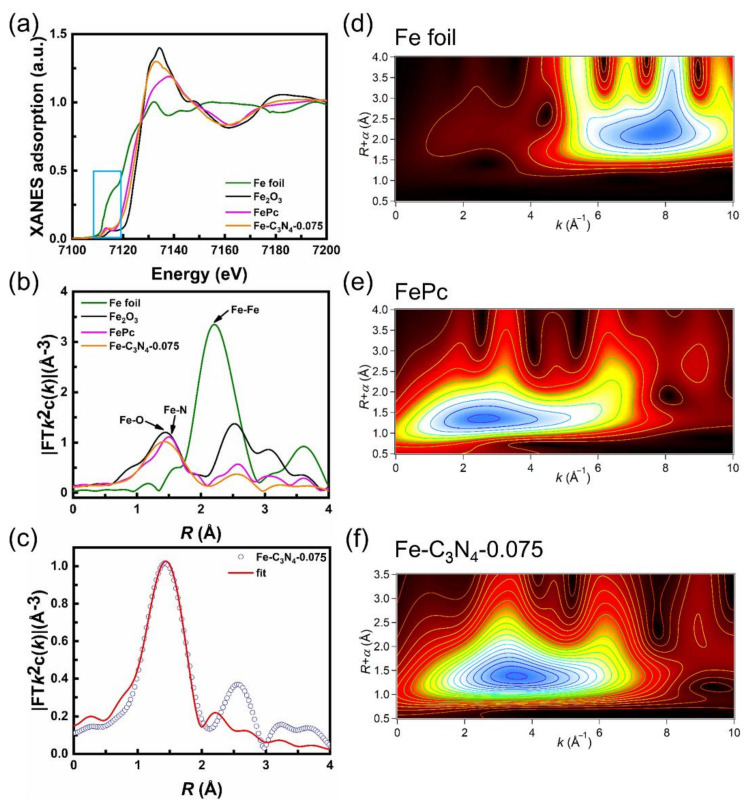
(**a**) XANES spectra (the orange area highlights the near-edge absorption energy). (**b**) Fourier transform (FT) of the Fe K-edge. (**c**) The corresponding EXAFS r space fitting curves of Fe-C_3_N_4_-0.075. Wavelet transform (WT) of Fe K-edge for (**d**) Fe foil, (**e**) FePc and (**f**) Fe-C_3_N_4_-0.075.

**Figure 6 nanomaterials-12-01416-f006:**
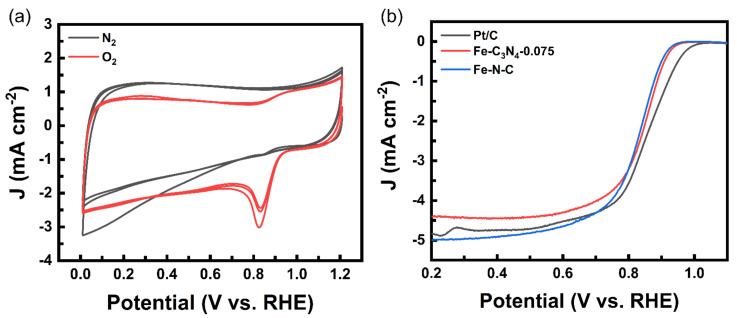
(**a**) CV curves of Fe-C_3_N_4_-0.075 in O_2_-saturated 0.1 M KOH with a sweep rate of 50 mV/s. (**b**) ORR polarization plots of Fe-C_3_N_4_-0.075 and Pt/C in O_2_-saturated 0.1 M KOH with a sweep rate of 5 mV/s and 1600 rpm.

**Table 1 nanomaterials-12-01416-t001:** Fitting results of Fe-C_3_N_4_-0.075 EXAFS.

Sample	Shell	N	R	σ^2^	R Factor (%)
Fe-C_3_N_4_-0.075	Fe-N	4.6 ± 0.2	2.00	0.009	0.37

Fitting range: Fe-C_3_N_4_-0.075:2.71 ≤ k (Å^−1^) ≤ 8.98 and 1.00 ≤ R (Å) ≤ 2.00.

## Data Availability

Not applicable.

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
