# Peer review of "Iron Single Atoms Anchored on Carbon Matrix/g-C3N4 Hybrid Supports by Single-Atom Migration-Trapping Based on MOF Pyrolysis"

_nanomaterials, 2022, doi:10.3390/nano12091416_

Round 1
Reviewer 1 Report
The manuscript titled “Iron single atoms anchored on carbon matrix/g-C3N4 hybrid supports by single-atom migration-trapping based on MOF pyrolysis” presented by Yining Jia , Rong Huang, and Ruijuan Qi deals with the study of single-atom migration-trapping for preparation of single-atom active centers of catalysts. However, there are minor issues that could be clarified:
- Line 23: SACs – what is that? Single-atom catalysts or typo?
- Whole manuscript: The absence of the space between word and [citations]. For example, lines 27-29.
- Line 203: “bond lengths for Fe‐N was 2.01 Å‐1”. Length is measured in angstroms.
The choice of methods to study the objects is adequate. To improve the scientific volume of the manuscript it is better to add the BET measurements surface specific area of synthetized samples (but not necessary for current form of manuscript. The XANES and EXAFS study of the Fe‐C3N4‐0.12 could compare both samples in terms of creating SA centers (if this data exist they should be added for discussion).
Reviewer 2 Report
The core of this work is to demonstrate that the migration-trapping methodology based on MOF pyrolysis works efficiently to generate iron single atoms on carbon matrix/g‐C3N4 hybrid support. Overall, this work is quite interesting and the manuscript is well organized. The title reflects the contents and the literature review is informative. Literature review part could also contain information about main principles of oxygen electroreduction on Fe single atom catalysts.
Physical characterization of the materials has been well interpreted and supports the conclusions drawn, which are sound and justified. The quality of the presentation is suitable, but I suggest that the manuscript needs a round of English editing. I found a number of grammar mistakes throughout the manuscript and suggest that the authors take a bit more care with editing.
While the work is attractive, there are some points that require further explanations and refinement:
-The loading of commercial Pt/C benchmark should be presented in experimental part.
-Main ORR kinetic parameters should be added to abstract section.
-Additionaly, BET measurements are essential when dealing with porous electrocatalyst materials.
-How about the electrochemical surface area (ECSA)? Please discuss it.
- What could be the reason for a large deviation of the diffusion limited current density values from the theoretical one for a 4-electron reduction of O2 in 0.1 M KOH (6 mA cm-2 at 1600 rpm)? Diffusion-limited current density obtained for a Pt/C catalyst is too low.
It is my opinion that there is enough new content provided in this communication to warrant publication in Nanomaterials, after minor revision.
